palaeontology/genetics

zooarchaeology, animal bones, bioarchaeology, ancient DNA, museum, petrous bone

**Author for correspondence:**
Albína Hulda Pálsdóttir
e-mail: albinap@gmail.com

# Not a limitless resource: ethics and guidelines for destructive sampling of archaeofaunal remains

Albína Hulda Pálsdóttir[1,2], Auli Bläuer[3,4], Eve Rannamäe[3,5], Sanne Boessenkool[1] and Jón Hallsteinn Hallsson[2]

[1]Centre for Ecological and Evolutionary Synthesis (CEES), Department of Biosciences, University of Oslo, Postbox 1066, Blindern, 0316 Oslo, Norway
[2]Faculty of Agricultural and Environmental Sciences, The Agricultural University of Iceland, Keldnaholti - Árleyni 22, 112 Reykjavík, Iceland
[3]Natural Resources Institute Finland (Luke), Itäinen Pitkäkatu 4 A, 20520 Turku, Finland
[4]University of Turku, Archaeology, Akatemiankatu 1, FI-20500 Turku, Finland
[5]Institute of History and Archaeology, University of Tartu, Jakobi 2, 51005 Tartu, Estonia

AHP, 0000-0002-5290-6862; AB, 0000-0003-0693-3373; ER, 0000-0003-1186-5512; SB, 0000-0001-8033-1165; JHH, 0000-0002-9127-2137

With the advent of ancient DNA, as well as other methods such as isotope analysis, destructive sampling of archaeofaunal remains has increased much faster than the effort to collect and curate them. While there has been considerable discussion regarding the ethics of destructive sampling and analysis of human remains, this dialogue has not extended to archaeofaunal material. Here we address this gap and discuss the ethical challenges surrounding destructive sampling of materials from archaeofaunal collections. We suggest ways of mitigating the negative aspects of destructive sampling and present step-by-step guidelines aimed at relevant stakeholders, including scientists, holding institutions and scientific journals. Our suggestions are in most cases easily implemented without significant increases in project costs, but with clear long-term benefits in the preservation and use of zooarchaeological material.

## 1. Introduction

The rapid advances in methods applied in archaeological research have been described as a 'third science revolution' [1]. The revolution encompasses many exciting new tools for zooarchaeological research, including quantitative modelling, ancient DNA (aDNA), isotopes and geometric morphometrics (e.g. [1–6]). These new methods can help answer questions that archaeologists and zooarchaeologists have been asking for decades [1,7–9].

Archaeofaunal remains are animal bones, teeth and mollusc shells found in human-associated contexts during archaeological excavations [10]. In some cases, other animal remains such as horn, skin, wool, hair and arthropods can be found, but these are rarer and usually need special burial environments such as very wet or very dry conditions [10]. Archaeofaunal remains formed through human activity rather than by natural events and the information value of archaeofaunal remains is highly connected to the human activities that formed them, such as hunting, rearing of livestock, butchery and production [9,10].

Traditional methods for analyses of archaeofaunal remains, such as speciation, measurement of bones and teeth, recording of pathology, taphonomy, wear stage and butchery marks, are in most cases non-destructive [10–14]. However, many of the newer tools such as aDNA, palaeoproteomics, collagen fingerprinting (or zooarchaeology by mass spectrometry (ZooMS)) and isotope analyses, require drilling into or cutting part of the sample specimen [6,15–18]. Such destructive sampling of animal bones and teeth from archaeological contexts is becoming more frequent with the increasing use of these methods, as evident by the rise in publications of isotope and aDNA studies (e.g. [1,6,18–25]). Archaeofaunal material extends the time series of samples available for studies of past animal populations back tens of thousands of years (e.g. [23,26–28]), and as more researchers outside of archaeology see this potential, the rate of destructive sampling is likely to continue to rise.

The currently available zooarchaeological collections have been accumulating since the nineteenth century, and at first glance, animal remains from archaeological sites may seem abundant, but at closer inspection it is evident they are a limited resource [29]. For most time periods and regions, the number of bones and teeth recovered from each species is surprisingly small despite large total quantities of material. Moreover, the material is unevenly divided between geographical locations and time periods due to variation in, for instance, preservation, settlement patterns, site history, recovery practices and the focus of the archaeological research [30]. When sampling for aDNA, isotopes or radiocarbon dating, it is important that samples come from different individuals and this considerably decreases the number of specimens available for sampling from each site. For example, in an urban archaeofaunal assemblage in Turku, Finland, there were only 20 horse (*Equus caballus*) elements among the 1300 kg of bones and teeth recovered, and these could anatomically have come from only two individuals [31]. At the early Anglo-Saxon (*ca* 450–700 CE) site at West Stow in the UK, a single brown bear (*Ursus arctos*) metacarpus was found in a collection of over 176 000 elements [32]. A recent study of cats from archaeological sites in Denmark found 30 cat specimens spanning the time period from 1100 BCE to 1000 CE, opposed to a much higher number of 3485 specimens from the period 1000–1660 CE [33], illustrating the often uneven distribution of finds between time periods. The number of bones from a species found at a site can also be misleading as often only a fraction come from well-dated contexts with clear taphonomic histories [34]. Finding bones and teeth from rarer species, e.g. microfauna and carnivores, is often only possible in large well-excavated archaeofaunal collections which have gone through zooarchaeological analysis and have been well curated (e.g. [34]). Such high-quality recovery and analyses are both expensive and time consuming [7,10,32,34].

In addition to the resource being limited, archaeofaunal collections are often poorly catalogued in holding institutions (e.g. [34,35]; AHP 2008–2018, AB 2004–2018, ER 2013–2018, personal observation). Moreover, the tendering of archaeological excavations due to construction, which is common today in many countries, obscures the overview of the composition of zooarchaeological collections, as many analyses are only published in grey literature reports that are often hard to find and access (e.g. [36–38]). The discarding of archaeofaunal collections without any analysis has been, and still is, common practice and it is not unusual that even analysed collections are lost due to budget cuts in museums or other holding institutions, conflict or natural disasters (e.g. [35,39–44]; AHP 2007–2018, AB 2004–2018, personal observation). This is not well communicated in the zooarchaeological literature and the perception of bountiful animal remains is reflected in statements such as: 'because at many important sites zooarchaeological material can be obtained in relatively large quantities across many stratigraphic layers' [5] and 'animal bones are more plentiful than human bones and represent all time periods and regions, allowing archaeogeneticists to refine their methods and test their assumptions about evolutionary genetics' [45]. These are just two examples, but the sentiment is common in discussions of archaeological animal remains and methods which require destructive analysis.

The rise in destructive sampling of archaeofaunal collections places holding institutions, such as museums and universities, in a difficult position. Getting funding for curation of collections is hard (e.g. [34,39,43]), and these institutions sometimes lack expertise both in zooarchaeology and the destructive methods. The combination of rapid methodological developments, rush for samples [46], and weak or absent rules or regulations governing destructive sampling of archaeofaunal material

(AHP 2013–2019, AB 2004–2018, ER 2011–2018, personal observation) places the burden of responsibility on those who grant the permits and select the samples. This situation raises new ethical dilemmas for stakeholders, including scientists, holding institutions and curators, journal editors and reviewers, who need to ensure that the limited, non-renewable and valuable resource of archaeofaunal remains is not used in ways which will negate its value to future research. Stakeholders cannot only rely on laws and guidelines and must critically examine research projects and consider ethical responsibilities towards heritage and future generations of scientists. Within the field of human osteology, the debate on ethical issues regarding sampling, research and display of archaeological human remains has been ongoing for decades and has become more vigorous with the recent increase in aDNA studies (e.g. [47–56]). However, to date there has been little discussion about the ethical implications of zooarchaeology as a field, the ethics of studying long-dead animals or the display of archaeofaunal remains (but see [57–59]). Although zooarchaeology can certainly use some of the lessons learned from the extensive literature on ethics in human osteology, including challenges regarding destructive sampling, proper documentation and sustainable use of limited material (e.g. [51–53]), many of the issues concerning human remains do not apply in the same way to animal remains. In some cases, the destructive sampling of non-human animal specimens is even seen as a less ethically and legally fraught alternative to the sampling of human remains, which is often governed by stricter laws, regulations and ethical guidelines [19,45,60]. Therefore, an independent discussion is needed within the field of zooarchaeology and among the different stakeholders, to fully develop an ethically sound approach to (destructive) archaeofaunal research.

Below we briefly explain the research potential of zooarchaeology and the significance of archaeofaunal collections, and why ethical considerations are called for. We then provide an overview of methods that require destructive sampling, and discuss the implications of the rush to sample petrous bones for ancient DNA. Finally, we suggest guidelines for scientists, holding institutions as well as journal editors and reviewers to mitigate the negative aspects of destructive sampling of archaeofaunal remains.

## 2. Research potential and significance of archaeofaunal collections

Zooarchaeology is a discipline that studies animal remains, most often animal bones and teeth from archaeological sites, with a strong focus on human–animal relationship in a broad sense [7,9,10]. A zooarchaeological collection and the single specimens within it may have a scientific, artistic and/or educational 'use value', an 'option value' to bridge 'knowledge gaps', as well as an 'existence value' relating to cultural identity [30,34]. Additionally, animal remains from archaeological contexts can be valuable for Indigenous communities [60], but this is rarely discussed. Animals have been an essential part of human society through history and their remains from archaeological excavations allow us to study the economies of the past, environmental change, evolution and human ritual behaviour, to name but a few of the common subjects of zooarchaeology [9,10].

Archaeofaunal collections do not stand alone; they form one part of the complex archive created during excavations, which includes artefacts, samples, measurements, drawings, photos and so forth [61]. Indeed, a large part of the value of an individual archaeofaunal item is its archaeological context and its connection to other material from the site [10,13,29,34]. The presence and abundance of animal species can be an indicator of past environmental changes, migrations and fluctuations in utilization of wild versus domestic resources [34]. Measurements of bones and teeth can be used to reconstruct the size and shape of the past individuals, reflecting domestication, adaptation and selection (e.g. [2,62,63]). Analysis of pathological changes are sources of information on animal use and abuse in the past (e.g. [64–68]). Butchery marks provide valuable data about cultural practices, tool use, preparation of meals and trade (e.g. [10,69,70]), and to understand depositional histories, the analysis of taphonomy is vital (e.g. [12,37,71]).

The duty to protect and preserve archaeological heritage is the fundamental principle of most ethical guidelines in archaeology (e.g. [29,72,73]). The work of zooarchaeologists is governed by many layers of conventions, laws, regulations and rules, which vary by region, country and institution. In 2009, the International Council for Archaeozoology (ICAZ) approved professional protocols which state: 'Clearly both the letter and the intent of all requirements pertaining to antiquities should be obeyed. Archaeozoologists must respect the laws of the countries in which they work, as well as respect general ethical principles of scholarship and research, both with regards to archaeozoological materials and to modern reference materials.' [29, p. 6]. However, rules and laws cannot address every conceivable scenario, some may actually conflict with each other and to practise ethical science, more than just compliance is needed [74,75].

# 3. Destructive sampling of archaeofaunal material

Destructive sampling is typically only briefly discussed in the zooarchaeological literature (e.g. [7,10,13,15,18]). The ICAZ professional protocols offer guidance on destructive sampling of archaeological animal bones and teeth: 'Destructive analysis (e.g. AMS dating, genetic analysis and isotope analysis) may provide information that is not otherwise available; however, decisions about destructive analysis should be taken carefully with the likely benefits balanced against the loss of the material, and consideration as to whether the information might be obtained without destruction or at a future time. The process should be carefully documented.' [29, p. 5]. Further, the Archaeozoology, Genetics, Proteomics and Morphometrics (AGPM) working group of ICAZ aims to facilitate communication between zooarchaeologists and molecular biologists to prevent 'uncoordinated multiplication of destructive sampling' [76]. This clearly illustrates a concern within the zooarchaeological community regarding destructive sampling, but the discussion has not progressed further or made its way into the many fields that are now sampling archaeofaunal remains, sometimes without a clear understanding of the value of the material.

Analyses requiring destructive sampling can give valuable insights into important research fields, such as domestication, development of livestock breeds, ritual activity and trade, and contribute to the preservation of local livestock breeds and endangered species (e.g. [5,6,21,23,26,77–83]). However, these analyses are most fruitful when done on archaeofaunal collections which have already been analysed with traditional zooarchaeological methods [7,23,32,45]. Taphonomic processes often complicate the stratigraphy of archaeological sites which in turn makes the dating of individual animal bones and teeth challenging [34]. Since the interpretation of data generated by aDNA or isotope studies relies heavily on the dating of a sample and its archaeological context, careful sample selection, that takes confounding factors such as taphonomy and stratigraphy into account, is needed to produce meaningful results [18,23,34,45].

Although many of the new methods (and some older ones like radiocarbon dating) can answer a multitude of exciting questions that traditional zooarchaeological methods cannot, they require cutting, drilling, scraping or chipping off chunks from the bone or tooth selected for sampling. The sample needed is usually small in weight—between 0.1 and 3 g—which typically results in damaging an area of up to $10 \times 10$ mm in size [15]. For small elements of medium-sized mammals (e.g. dogs, sheep, pig) or small animals (e.g. fish, birds, rodents) sampling can use up whole bones, especially when multiple methods like aDNA and radiocarbon dating are being used on the same specimen. Fortunately, methodological advancements are leading to ever decreasing samples sizes, even for established methods like radiocarbon dating (e.g. [84]). Efforts are also being made to enable the use of a single sample for both aDNA extraction and radiocarbon dating [85], and ZooMS has been introduced as a less destructive and cheaper method to test for the presence of collagen and taxonomic determination that can be performed before larger samples are taken for further analyses (e.g. [86,87]).

Any destructive sampling can lessen the value of the specimen for future research and/or display [15,29,53]. Some measurements can no longer be taken after sampling and surface modifications such as cut marks can be obscured. Additionally, bones and teeth can break in unintended ways during sampling even when it is meant to be minimally damaging. Moreover, while destructive sampling can enhance the importance of a collection to the researchers by providing data about the past, it can diminish or change its value for Indigenous groups (e.g. [18,48,58,60]). These competing interests and values must be considered and reconciled before destructive sampling.

Not all destructive sampling yields usable results. The success rates of destructive methods vary greatly, depending on the preservation of the material sampled and methods used. Radiocarbon dating, isotope analysis and ZooMS generally have a high rate of success, but for palaeoproteomics and aDNA it is common for samples to fail (e.g. [17,86,88,89]). Even though much effort has been spent on trying to predict aDNA preservation in archaeological material and the rate of DNA degradation in ancient samples, it still remains poorly understood and it is not possible to know with certainty whether the analysis of a particular sample will be successful (e.g. [90–98]).

Research has shown that DNA continues to degrade after excavation and the length of time since excavation negatively impacts the preservation of DNA [96,99]. Climate controlled and low oxygen storage facilities can slow this process, but such facilities are expensive to build and maintain, and archaeofaunas are often stored in buildings which fluctuate in both heat and humidity to the detriment of DNA preservation ([45,100]; AHP 2005–2018, AB 2004–2018, ER 2012–2018, personal observation). Additionally, faunal collections can be damaged, lost, deaccessioned or discarded after being excavated. Therefore, not allowing destructive sampling or postponing it until 'later' when methods have been further refined does in some cases lead to a loss of value and a lost opportunity for greater understanding of the past.

# 4. The special case of the petrous bone for aDNA analyses

The petrous bone is located in the temporal region of mammalian skulls and is the densest bone in the skeleton [101]. Petrous bones have become the first choice for aDNA sampling after they were shown to have better DNA preservation compared to other bones, and pressure to sample petrous bones has accordingly increased significantly [46,47,49,93,102]. In human burials, petrous bones from both sides of the skull may be recovered, allowing researchers to sample only one of them and leaving the other untouched [54,102–104]. Unfortunately, most archaeofaunal collections do not include many complete skulls and even in large assemblages there is a limited number of skull fragments, and only a subset of them will have the petrous intact ([105]; AHP 2005–2018, AB 2004–2018, ER 2017–2018, personal observation). Compounding the problem, animal petrous bones are infrequently identified to species and are often not recorded at all during traditional zooarchaeological analysis despite generally preserving well due to their density [105–109]. Therefore, there is typically little information available about the frequency of petrous bones in zooarchaeological collections ([105]; AHP 2005–2018, AB 2004–2018, ER 2017–2018, personal observation).

Human petrous bones have been used for fetal ageing [110], sexing [111] and for dietary isotope studies [112] in addition to their value for aDNA research (e.g. [93,102]). In animals, however, little is known about the research potential of petrous bones apart from their value for aDNA. There are indications that non-destructive analysis of animal petrous bones could be used to study domestication and ageing, and to differentiate between sheep breeds [108,113]. Since there are almost no standards on how to record animal petrous bones, it is difficult to ensure that sufficient information is recorded before sampling takes place.

Petrous bones from different species vary in size, which affects sampling decisions. The petrous of a sheep, goat or a pig is for instance approximately $0.5 \times 1.0 \times 2.0$ cm in size, cattle petrous is only slightly larger and horse petrous about twice that size. The size of a petrous bone dictates the number of potential samples that can be taken, and the petrous bones of many common domesticates can realistically only be sampled once after their surface has been cleaned. Selecting the densest part of the petrous bone for maximum recovery of endogenous DNA and minimizing damage to the sample is also challenging and requires practice ([54,114,115]; personal observation). Although sampling methods are constantly being improved to reduce sample damage and maximize data recovery, these efforts primarily focus on the human petrous and they are not always applicable to animal petrous bones due to anatomical differences [54,104,109,116].

# 5. What can be done? Guidelines for stakeholders

There are many ways to minimize the negative effects of destructive sampling on archaeofaunal collections. Below we introduce a set of guidelines aimed at different groups of stakeholders. Although many of these guidelines may appear self-evident, in practice they are often not or only partly applied. They can serve as a starting point for further dialogue among the different parties involved.

## 5.1. Researchers sampling from archaeofaunal collections

Many types of projects initiated by scientists from such diverse fields as genomics, biology, geology and archaeology are now doing destructive sampling on archaeofaunal material. This section focuses on best practices for scientists planning and performing this sampling.

(1) *Collaborate*: Finding the right samples requires specialized knowledge of a region and its archaeology and is a vital step for any research project using destructive sampling. Involve an archaeologist and/or zooarchaeologist in the project from the start if possible. A zooarchaeologist can assist with sample selection, identify the best part of the specimen to sample for maximum results and minimize damage to diagnostic and taphonomically important parts of a bone or tooth. Archaeologists familiar with a site can assist with clarifying stratification and dating of samples as well as interpretation of results. It may therefore be appropriate for archaeologists or zooarchaeologists working in a region and/or time period to be full participants in the research, including participation in interpretation of data and writing of articles for publication following the Vancouver guidelines [117].

(2) *Have a sampling strategy*: What species, element, time period and type of site are you looking for and why? The 'ideal sample' comes from a clear well-dated archaeological context. Different methods

require different samples, e.g. petrous for aDNA, tooth enamel for strontium isotopes. Make sure you are not sampling the same individual more than once, for example, by sampling only specific elements from either the right or left side, and/or elements from separate contexts [54,100].

(3) *Know your samples*: Ensure to have information on the archaeological context, date, time since excavation, cleaning, conservation and storage. Often this information can only be found in institution archives. It is advisable to collect this information when the sample is selected.

(4) *Permissions*: Obtain written permissions from all relevant parties. Make sure that all parties understand what sampling is being permitted and be clear about how long these permissions will last. Ask if permissions need to be renewed for other analyses or new projects. Make an agreement with the holding institution at the time of sample selection what will be returned (bones, teeth, powder, derived products) and when. In many cases any leftover bone or tooth powder, isolated collagen, DNA extracts and other sample products will be best kept in the laboratory that performed the analysis, as holding institutions often do not have the facilities to store the material. Return samples and leftovers in a timely manner once the project is completed.

(5) *Documentation*: Document the specimen in detail before sampling. Take standard measurements, record weight and take high-quality photos from multiple angles with scale and sample information. This data should be shared with excavation directors and/or holding institutions. Ideally, measurements should be published in supplements of articles stemming from the project. Describe sampling techniques so others can learn from your experience.

(6) *Sample size*: Take only what you need, but take enough to maximize the likelihood of usable results. Document the remaining specimen and share this information with the holding institution. Conduct pilot studies with a small number of lower value samples to test and adjust methods where possible. Be aware of the best practices; within rapidly developing fields like aDNA and proteomics a continual adjustment of sampling strategy and laboratory methods based on the latest research is necessary, especially because new methods are often less destructive and yield more and better quality data. Use a single sample for multiple analyses when possible (e.g. [85,87]).

(7) *Traceability*: Ensure it is easy to trace each sample from your published papers and raw data back to the holding institution, site publications, archaeological context and specialist reports. This makes it easier for others to use generated data, enhances reproducibility and prevents unnecessary repetitions of destructive sampling.

(8) *Know what you have in your laboratory*: Incoming samples should be registered immediately with all necessary information to ensure identification and traceability. If leftovers (powder, sampling products etc.) remain in the laboratory when a project has finished, an inventory of this material needs to be kept. Once a project is completed give other researchers access to the material if requested, but ensure that permissions from the original holder of the material are obtained.

(9) *Share generated data*: Share raw data in public databases with clear meta-data for each sample that can be connected to publications and holding institutions. The holding institution should receive copies of all publications that use samples or data derived from the institution's collection.

(10) *Publish negative results*: Sharing what you tried and did not work will prevent unnecessary destruction of valuable samples by other researchers. This can be done as part of the publication of positive results, by publishing a sampling report (e.g. [118]) or in dedicated journals.

## 5.2. Institutions responsible for archaeofaunal collections

To make sound decisions regarding destructive sampling, holding institutions need to have a complete overview of their zooarchaeological collections and the methods used to study them. Unfortunately, with funding pressure and reduction in staff prevalent in the museum industry today, curatorial knowledge of collections is threatened (e.g. [39,41,43,119]). The increasing number of requests for destructive analysis adds to the challenges faced by holding institutions [47]. Ideally the holding institution is a participant in the research project but due to time and funding restraints that is not always possible.

(1) *Create a paper trail*: Have a standard form that is filled out by those requesting to do destructive sampling. This will make it easier to keep track of requests and standardize the information collected about sampling requests.

(2) *Consult with specialists*: The abundance of archaeofaunal material varies between time periods and locations. Having some idea about the abundance of the available material is important

when reviewing applications for destructive analysis. If this knowledge is not available within the institution, it is advisable to consult with an outside zooarchaeologist before giving the sampling permission.

(3) *Know the project*: The researchers involved should have the necessary experience to complete the study, and projects should ideally include archaeologists with knowledge of the sampled material, the region and/or time period. Ask where the technical work will take place and what kind of facilities are needed, e.g. for aDNA and palaeoproteomics a dedicated clean laboratory is needed for sample processing. Ask for information about funding for the planned research: applicants should be able to demonstrate that they have adequate funding to complete the project. Verify that the specimens to be sampled can answer the questions the project aims to answer. The goal of a project and why it requires destructive sampling should be clear.

(4) *Sample size*: How big is the sample required and how will it be taken? Ideally there should be an open dialogue about sample selection, sample sizes and sampling locations between the researchers who sample the material and those responsible for its curation. While aDNA can be done on as little as 50 mg of powder, it is often necessary to remove more material to clean the surface of the sample and get to the part that is preferred for sampling. A common sample size for an aDNA extraction is 100–400 mg (e.g. [21,54,83,120]). Holding institutions should encourage integrating the use of multiple methods on a single sample when possible (e.g. [85,87]).

(5) *Documentation*: Require detailed documentation before sampling. At a minimum this should include species and skeletal element ID, measurements (standard zooarchaeological measurements, weight and overall size) and multiple photographs with scale. The writing of sampling reports which include information recorded about the sample, photos from before and after sampling and methodology can be a good way to make sure this information is accessible.

(6) *State clearly what should be returned*: This will vary between projects, but it should be clear from the beginning. Any bone or tooth fragment bigger than 1–2 g should generally be returned. In many cases any leftover bone or tooth powder, isolated collagen, DNA extracts and other sample products will be best kept in the laboratory that performed the analysis, as holding institutions often do not have the expertise or facilities to store the material. State clearly how leftovers can be used or if new permission should be sought. Holding institutions should encourage analyses of leftover material when suitable.

(7) *Data*: Require access to resulting data. Decide what data and in which format has to be made available by the researcher. Some data is best shared in public databases (e.g. genomic data where the raw datasets are very large), some in data publications and some can be reported in sampling reports. All data generated should be easily traced back to the original sample and archaeological context.

(8) *Acknowledgement*: Require clear acknowledgement of the holding institution in all publications. The holding institution should receive copies of all publications that use samples or data derived from the institution's collection. This can help justify funding for curation and storage facilities.

(9) *Saying no*: Sometimes holding institutions need to decline requests for destructive sampling, e.g. when proposed projects cannot meet sampling guidelines or the requested material is unsuitable for destructive analysis. Rejections should be given in a timely manner and state clearly why the request was declined. In some cases, alternative samples or sampling strategies can be suggested which would still allow a project to reach most of its goals.

(10) *Share results*: Use research results to promote collections. Link the publications to institution's social media and exhibitions, ask researchers to give public lectures.

## 5.3. Journals publishing studies that include samples from archaeofaunal collections

Journal editors, editorial boards and reviewers are of great importance in upholding ethical standards in science and are therefore imperative actors in controlling the ethical use of samples from archaeofaunal collections. Many papers on zooarchaeological material are published in multidisciplinary journals and editors cannot be expected to have experience with the ethical issues.

(1) *Citations*: Require citation of the relevant archaeological literature. It is vital that samples can be traced back to the original excavations and placed in the correct context.

(2) *Traceability*: Require that specimen or collection numbers for each sample are included along with references to the original site and specialist reports when available. These can be museum catalogue numbers or site-specific numbers that can be linked back to the original excavation

archive. Numbers assigned by laboratories that preform destructive analysis usually cannot be directly traced back to archaeological context so they are not sufficient.

(3) *Data accessibility*: Raw data resulting from genomics and proteomics analysis should always be made available in data depositories so they are available for re-analyses. Deposition of raw data from other destructive analyses in open access depositories should be encouraged when possible.

(4) *Acknowledgement*: Require the holding institutions to be acknowledged. This serves both to emphasize their important role in curating and protecting collections and makes it easier for them to demonstrate research on their collections.

(5) *Consider the author list*: Finding the right samples requires specialized knowledge of a region and its archaeology and is a vital step for any research project using destructive sampling. It may therefore be appropriate for archaeologists or zooarchaeologists working in a region and/or time period to be full participants in the research, including participation in interpretation of data and writing of articles for publication following the Vancouver guidelines [117].

(6) *Ethical declarations*: Journal editors need to ask for a declaration from article authors that all samples were taken with permission, according to local laws and regulations, and in an ethically responsible manner. This will help encourage scientists to think about the ethics behind sampling archaeofaunal material.

# 6. Conclusion

Developing methods such as three-dimensional photography, models and accessible CT scanning of bones and teeth before sampling will in the future be useful ways of recording specimens before destructive sampling. However, no methodological advancement can completely remove ethical considerations when it comes to destructive sampling of archaeofaunal material. Animal bones and teeth from archaeological sites are not an unlimited resource, and as part of human cultural heritage their rarity and uniqueness needs to be understood and respected by those sampling them. The ethical issues regarding the importance of archaeofaunal material to Indigenous populations have not been addressed here as they largely fall outside of our scope of experience. However, when reviewing the zooarchaeological literature, it is clear that such discussion is needed.

The practice of zooarchaeology, and specifically destructive sampling, is clearly not without ethical implications and open discussion within the field is vitally important. With many different stakeholders such discussion needs to take place in the form of an open dialogue with museums, holding institutions, archaeologists, molecular biologists, geologists and others involved in destructive sampling of archaeofaunal material. The takeaway from this paper should definitely not be that we should stop sampling while we wait for methodological improvements or even that we should be sampling less, but that we need to make sure that destructive sampling is done in a conscious and responsible manner. By working together, zooarchaeologists, holding institutions and other researchers can protect archaeofaunal collections as a valuable resource for future research and an important part of human heritage.

Ethics. This research used no human or animal subjects or sampling of archaeological material.

Data accessibility. No datasets are associated with this research.

Authors' contributions. A.H.P. initiated the work which was then developed and written with A.B., E.R., S.B. and J.H.H. All authors gave final approval for publication.

Competing interests. The authors declare no conflicting interests.

Funding. This work was supported by the Icelandic Research Fund grant no. 162783-051, Finnish Academy grant no. SA286499, the European Union's Horizon 2020 research and innovation programme under the Marie Skłodowska-Curie grant agreement no. 749226 and Estonian Research Council grant nos. PRG29 and IUT 20-7.

Acknowledgements. We acknowledge Anna Drożdżowicz for valuable comments on early versions of the manuscript. We thank the two reviewers and the associate editor for their valuable input on the paper.

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
