## [Reviewer comments · Royal Society Open Science]

Review History

RSOS-191059.R0 (Original submission)

Review form: Reviewer 1 (Marjan Mashkour)

Is the manuscript scientifically sound in its present form?

Yes

Are the interpretations and conclusions justified by the results?

Yes

Is the language acceptable?

Yes

Do you have any ethical concerns with this paper?

No

Have you any concerns about statistical analyses in this paper?

No

Recommendation?

Accept with minor revision (please list in comments)

Comments to the Author(s)

The paper by Albína Hulda Pálsdóttir and collaborators: Not a limitless resource ... » is a reflection that falls just on time and somehow simultaneous to the environmental crisis ongoing debates.

This is one of the more sound useful and complete updates I have read on the question of the preservation of archaeofaunal material. The paper is presenting first the general picture of the pivotal role of bioarchaeological remains, including human and animals remains, in ancient molecular fields, namely genetics and biogeochemistry. The authors are very right to highlight that despite some reflection on the use of human remains, undoubtedly because of underlying ethical issues, the same views have been much less followed for animals. I would also add to this the plant remains that is, in parallel, experiencing destructive analysis pressures more and more.

Today people curating bioarchaeological material are facing a critical transitional period of how to preserve these remains and in the meantime to use or to give access to the material in order to promote scientific research questions.

I have no reservation for the publication of this paper that I highly support. It should be translated in the language of different countries holding archaeofaunal or osteological remains in their Museums, that will allow a more accessible understanding of the contents and its application. This paper that is a very well structured vade-mecum for having an ethical and responsible attitude towards archaeofauna, exactly like for any other category of cultural heritage materials.

Comments and further reading suggestions

-Line 56 : To add to the references quoted regarding grey literature reports: Cécile Callou, Isabelle Baly, Olivier Gargominy, Elodie Rieb. National Inventory of Natural Heritage website : recent, historical and archaeological data. SAA Archaeological Record, Society for American Archaeology (SAA), 2011, 11 (1), pp.37-40.

-Line 59 the experience of partial recovery of the osteological collections at the National Museum of Iran after the Iranian Revolution and further political upheavals is a very appropriate example on the debate about discarded archaeofaunal assemblages : Mashkour M. Biglari, F. and Ghafouri N. with the collaboration of Ahadi G., Amiri S., Beyzaei Doust S., Davoudi H., Fathi H., Khazaeli R., Moradi B., Yashmi R. 2012. The OsteoArchaeological Project of the National Museum of Iran, an Interim Report. Journal of Iranian Archaeology, 3: 72-76.

- Line 172 the authors could add Bollongino R. & Vigne J.-D., 2008.- Temperature monitoring in archaeological animal bone samples in the Near East arid area, before, during and after excavation. J. Archaeol. Sci., 35 : 873-881. The publications address exactly the question of DNA loss after excavation.

Part 5.1 I would like to insist on the pivotal involvement of the primary actors: archaeologist without whom archaeozoologist could never have access to the material and the archaeozoologist without whom the geneticists or other biologists could not do there research. These people should not be only thanked, but should be fully part of the project, not only as sample provider but taking part in the research and certainly co-authors. I would really appreciate if this case be more clearly expressed.

As for the Institutions, it is of course very important for them to take fully part to the outcomes of the research, not as an option but an obligation, since the Institution, generally public, so countries basically, have invested a huge amount of budget for the excavations to exist. And even when the country itself is not the investor, it still has the full authority on it own country's

biological heritage. The accountability and transparency attitudes are often not understood or adopted and need to be reminded clearly.

Finally I would add my personal experience regarding the traceability of the samples. The archaeozoologist should make sure there is an agreement between the labs to use similar codes and to make sure that the results of the raw data be published with the initial codes of the archaeozoologist (with the whole archaeological context data) and the next lab's codes. This is also a crucial step for not losing track of the archaeological information.

Review form: Reviewer 2 (Terry O'Connor)

Is the manuscript scientifically sound in its present form?

Yes

Are the interpretations and conclusions justified by the results?

Yes

Is the language acceptable?

Yes

Do you have any ethical concerns with this paper?

No

Have you any concerns about statistical analyses in this paper?

No

Recommendation?

Accept with minor revision (please list in comments)

Comments to the Author(s)

This is a well-timed and necessary review of the subject, with clear proposed guidelines. In particular, the paper doesn't simply say "Do this" but says "Do this because...". The language is clear and concise.

I have a few specific comments and suggestions.

21: for 'crustaceans', put 'arthropods' so this includes the mites and insects that can be very abundant in anoxic sediments.

35-52: the point about scarcity is well made here.

Introduction - very good. The issue at hand is set out clearly, with enough supporting detail but without over-stating the case. 88-92 then leads into the rest of the paper.

2 Research Potential... this is good as it stands. The text mentions the importance of context.

Maybe this could be linked back to the issue of specimen scarcity: i.e. even when one has extracted the 35 cats from a tonne of bones, only some of them will have good stratigraphic and dating context.

135-8: context again - in my experience, too many of those undertaking destructive analysis have a poor understanding of the taphonomic processes that underlie (and screw up!) stratigraphic and dating context, resulting in material of poor strat integrity being used naively. You make this point briefly at 143-5, but maybe it could be made more strongly?

155-6: for 'species determination' put 'taxonomic determination'. I know this is pernickety of me, but ZooMS only occasionally gives sound results to species level, more usually genus or higher.

204-5: 'they do not always apply to...'. I'm not entirely clear on the intended meaning. Do you mean that sampling standards for human petrous bones *are not* applied to other species, or that they *can not* be applied because of differences in anatomy? If the latter, then 'they are not always applicable to...' might be better?

5.1 – this is good. These guidelines are essentially only 'best practice', but experience shows how necessary it is to have them laid out like this. In particular, publication of negative results is important and easily overlooked.

281-2: 'Verify that the specimens to be sampled can answer the questions the project aims to answer'. Again, this should be obvious, but it is so often not clearly the case.

Section 5.2 is clear and sensible. I would add one more guideline however: "Be prepared to say no. For destructive sampling, this should be the default answer unless other guidelines are clearly met". And 305-6 Acknowledgement should include the holding institution receiving copies of all published work that utilises the samples or data.

References: thorough, covers the key literature. All reviewers suggest adding one of their own papers. The following is relevant to the issue of quality assurance in very large bone collections: Rainsford, C., O'Connor, T., & Connelly, P. (2016). The embarrassment of riches: rationalising faunal assemblages from large urban sites. *International Journal of Osteoarchaeology*, 26(2), 221-231.

Decision letter (RSOS-191059.R0)

23-Aug-2019

Dear Ms Palsdottir

On behalf of the Editors, I am pleased to inform you that your Manuscript RSOS-191059 entitled "Not a limitless resource – Ethics and guidelines for destructive sampling of archaeofaunal remains" has been accepted for publication in Royal Society Open Science subject to minor revision in accordance with the referee suggestions. Please find the referees' comments at the end of this email. Note that the Associate Editor considers this to be a very timely paper - thank for choosing to publish it in Open Science

The reviewers and handling editors have recommended publication, but also suggest some minor revisions to your manuscript. Therefore, I invite you to respond to the comments and revise your manuscript.

- Ethics statement

- Data accessibility

It is a condition of publication that all supporting data are made available either as supplementary information or preferably in a suitable permanent repository. The data accessibility section should state where the article's supporting data can be accessed. This section should also include details, where possible of where to access other relevant research materials such as statistical tools, protocols, software etc can be accessed. If the data has been deposited in

an external repository this section should list the database, accession number and link to the DOI for all data from the article that has been made publicly available. Data sets that have been deposited in an external repository and have a DOI should also be appropriately cited in the manuscript and included in the reference list.

<http://datadryad.org/submit?journalID=RSOS&manu=RSOS-191059>

- **Competing interests**

- **Authors' contributions**

- **Acknowledgements**

- **Funding statement**

Because the schedule for publication is very tight, it is a condition of publication that you submit the revised version of your manuscript before 01-Sep-2019. Please note that the revision deadline will expire at 00.00am on this date. If you do not think you will be able to meet this date please let me know immediately.

on behalf of Professor Matthew Collins (Associate Editor) and Jon Blundy (Subject Editor)
 openscience@royalsociety.org

Associate Editor Comments to Author (Professor Matthew Collins):

I fully concur with referee 2 that ZooMS can generally not resolve to species level.

In recommendation to

5.1 Researchers sampling from archaeofaunal collections

I would encourage, although not demand, that a statement should be included encouraging integrated analysis - e.g. :-

“In many cases any leftover bone or tooth powder, isolated collagen, DNA extracts, and other sample products will be best kept in the lab that performed the analysis, as holding institutions often do not have the facilities to store the material.” Generally, more can be learnt by integrating analyses, researchers should consider and curators encourage the future additional analyses of left-over residues or their co-extraction at the time of analyses.

5.3 Journals publishing studies that include samples from archaeofaunal collections

I would suggest that publishers insist that in the case of complex datasets, e.g. (genomics / proteomics) raw data should be made available in a recognised data depository for subsequent re-analyses

Reviewer comments to Author:

Reviewer: 1

The paper by Albína Hulda Pálsdóttir and collaborators: Not a limitless resource ... » is a reflection that falls just on time and somehow simultaneous to the environmental crisis ongoing debates.

This is one of the more sound useful and complete updates I have read on the question of the preservation of archaeofaunal material. The paper is presenting first the general picture of the pivotal role of bioarchaeological remains, including human and animals remains, in ancient molecular fields, namely genetics and biogeochemistry. The authors are very right to highlight that despite some reflection on the use of human remains, undoubtedly because of underlying ethical issues, the same views have been much less followed for animals. I would also add to this the plant remains that is, in parallel, experiencing destructive analysis pressures more and more.

Today people curating bioarchaeological material are facing a critical transitional period of how

to preserve these remains and in the meantime to use or to give access to the material in order to promote scientific research questions.

I have no reservation for the publication of this paper that I highly support. It should be translated in the language of different countries holding archaeofaunal or osteological remains in their Museums, that will allow a more accessible understanding of the contents and its application. This paper that is a very well structured vade-mecum for having an ethical and responsible attitude towards archaeofauna, exactly like for any other category of cultural heritage materials.

Comments and further reading suggestions

-Line 56 : To add to the references quoted regarding grey literature reports: Cécile Callou, Isabelle Baly, Olivier Gargominy, Elodie Rieb. National Inventory of Natural Heritage website : recent, historical and archaeological data. SAA Archaeological Record, Society for American Archaeology (SAA), 2011, 11 (1), pp.37-40.

-Line 59 the experience of partial recovery of the osteological collections at the National Museum of Iran after the Iranian Revolution and further political upheavals is a very appropriate example on the debate about discarded archaeofaunal assemblages : Mashkour M. Biglari, F. and Ghafouri N. with the collaboration of Ahadi G., Amiri S., Beyzaei Doust S., Davoudi H., Fathi H., Khazaeli R., Moradi B., Yashmi R. 2012. The OsteoArchaeological Project of the National Museum of Iran, an Interim Report. Journal of Iranian Archaeology, 3: 72-76.

- Line 172 the authors could add Bollongino R. & Vigne J.-D., 2008.- Temperature monitoring in archaeological animal bone samples in the Near East arid area, before, during and after excavation. J. Archaeol. Sci., 35 : 873-881. The publications address exactly the question of DNA loss after excavation.

Part 5.1 I would like to insist on the pivotal involvement of the primary actors: archaeologist without whom archaeozoologist could never have access to the material and the archaeozoologist without whom the geneticists or other biologists could not do their research. These people should not be only thanked, but should be fully part of the project, not only as sample provider but taking part in the research and certainly co-authors. I would really appreciate if this case be more clearly expressed.

As for the Institutions, it is of course very important for them to take fully part to the outcomes of the research, not as an option but an obligation, since the Institution, generally public, so countries basically, have invested a huge amount of budget for the excavations to exist. And even when the country itself is not the investor, it still has the full authority on its own country's biological heritage. The accountability and transparency attitudes are often not understood or adopted and need to be reminded clearly.

Finally I would add my personal experience regarding the traceability of the samples. The archaeozoologist should make sure there is an agreement between the labs to use similar codes and to make sure that the results of the raw data be published with the initial codes of the archaeozoologist (with the whole archaeological context data) and the next lab's codes. This is also a crucial step for not losing track of the archaeological information.

Reviewer: 2

Comments to the Author(s)

This is a well-timed and necessary review of the subject, with clear proposed guidelines. In particular, the paper doesn't simply say "Do this" but says "Do this because...". The language is clear and concise.

I have a few specific comments and suggestions.

21: for 'crustaceans', put 'arthropods' so this includes the mites and insects that can be very abundant in anoxic sediments.

35-52: the point about scarcity is well made here.

Introduction – very good. The issue at hand is set out clearly, with enough supporting detail but without over-stating the case. 88-92 then leads into the rest of the paper.

2 Research Potential... this is good as it stands. The text mentions the importance of context.

Maybe this could be linked back to the issue of specimen scarcity: i.e. even when one has extracted the 35 cats from a tonne of bones, only some of them will have good stratigraphic and dating context.

135-8: context again – in my experience, too many of those undertaking destructive analysis have a poor understanding of the taphonomic processes that underlie (and screw up!) stratigraphic and dating context, resulting in material of poor strat integrity being used naively. You make this point briefly at 143-5, but maybe it could be made more strongly?

155-6: for 'species determination' put 'taxonomic determination'. I know this is pernickety of me, but ZooMS only occasionally gives sound results to species level, more usually genus or higher.

204-5: 'they do not always apply to...'. I'm not entirely clear on the intended meaning. Do you mean that sampling standards for human petrous bones *are not* applied to other species, or that they *can not* be applied because of differences in anatomy? If the latter, then 'they are not always applicable to...' might be better?

5.1 – this is good. These guidelines are essentially only 'best practice', but experience shows how necessary it is to have them laid out like this. In particular, publication of negative results is important and easily overlooked.

281-2: 'Verify that the specimens to be sampled can answer the questions the project aims to answer'. Again, this should be obvious, but it is so often not clearly the case.

Section 5.2 is clear and sensible. I would add one more guideline however: "Be prepared to say no. For destructive sampling, this should be the default answer unless other guidelines are clearly met". And 305-6 Acknowledgement should include the holding institution receiving copies of all published work that utilises the samples or data.

References: thorough, covers the key literature. All reviewers suggest adding one of their own papers. The following is relevant to the issue of quality assurance in very large bone collections: Rainsford, C., O'Connor, T., & Connelly, P. (2016). The embarrassment of riches: rationalising faunal assemblages from large urban sites. *International Journal of Osteoarchaeology*, 26(2), 221-231.

Author's Response to Decision Letter for (RSOS-191059.R0)

See Appendix A.

Decision letter (RSOS-191059.R1)

09-Sep-2019

Dear Ms Palsdottir,

I am pleased to inform you that your manuscript entitled "Not a limitless resource – Ethics and

guidelines for destructive sampling of archaeofaunal remains" is now accepted for publication in Royal Society Open Science.

on behalf of Professor Matthew Collins (Associate Editor) and Jon Blundy (Subject Editor)
openscience@royalsociety.org

Associate Editor Comments to Author (Professor Matthew Collins):

Associate Editor: 1

Comments to the Author:

(There are no comments.)

Reviewer comments to Author:

Follow Royal Society Publishing on Twitter: [@RSocPublishing](https://twitter.com/RSocPublishing)

Appendix A

Reykjavík, August 29th 2019

To the Editor,

We are pleased to re-submit our manuscript entitled “Not a limitless resource – Ethics and guidelines for destructive sampling of archaeofaunal remains” after revision for publication in *Royal Society Open Science* in the research article – communications category.

We found the suggestions of the editor and reviewers to be very relevant and helpful in improving the manuscript.

Response to editor’s comments

1. ZooMS ‘species determination’ changed to ‘taxonomic determination’ following the recommendation of reviewer 2.
2. Section 5.1 and 5.2 on integrated analysis and use of leftovers. Changes underlined:
 - a. 5.1 added at the end of no 6: “. Use a single sample for multiple analyses when possible [e.g. 85,87].”
 - b. 5.2 added at the end of no 4: “Holding institutions should encourage integrating the use of multiple methods on a single sample when possible [e.g. 85,87].”
 - c. 5.2 no 6: “State clearly how leftovers can be used or if new permission should be sought. Holding institutions should encourage additional analyses of leftover material when suitable.”
3. Section 5.3 no 3 changes underlined: “**Data accessibility: Raw data resulting from genomics and proteomics analysis should always be made available in data depositories so they are available for re-analyses. Deposition of raw data from other destructive analyses in open access depositories should be encouraged when possible.**”

Response to reviewer comments

Reviewer 1

1. Line 46: Reference Callou et al 2011 was added.
2. Line 59: Reference Mashkour et al 2012 was added.
3. Line 172: Reference Bollongion et al 2008 was added.

Postal address:
Department of Biosciences,
P.O.Box 1066 Blindern, N-0316 OSLO, Norway
Visiting address:
Kristine Bonnevis hus,
Blindernveien 31, N-0316 OSLO, Norway

Telephone: +47 22 85 56 00
Telefax: +47 22 85 47 26
E-mail: postmottak@ibv.uio.no
<http://www.mn.uio.no/ibv/english>
Org.nr.: 971 035 854

4. Part 5.1, we made some changes to address but feel that authorship is not in all cases appropriate for all archaeologists and zooarchaeologists that may participate in procuring samples but have added the Vancouver guidelines as in 5.3 no 5 which emphasize offering data providers the option to be full participants: “Collaborate: Finding the right samples requires specialised knowledge of a region and its archaeology and is a vital step for any research project using destructive sampling. Involve an archaeologist and/or zooarchaeologist in the project from the start if possible. A zooarchaeologist can assist with sample selection, identify the best part of the specimen to sample for maximum results, and minimize damage to diagnostic and taphonomically important parts of a bone or tooth. Archaeologists familiar with a site can assist with clarifying stratification and dating of samples as well as interpretation of results. It may therefore be appropriate for archaeologists or zooarchaeologists working in a region and/or time period to be full participants in the research, including participation in interpretation of data and writing of articles for publication following the Vancouver guidelines [117].”
5. Participation of holding institutions 5.2. To address this comment we added the underlined “The increasing number of requests for destructive analysis adds to the challenges faced by holding institutions [47]. Ideally the holding institution is a participant in the research project but due to time and funding restraints that is not always possible.” While the full participation of holding institutions in research is ideal it has been our experience that this is often not possible due how understaffed many museums and holding institutions are.
6. Traceability of the samples, change underlined for 5.1 no 7, “Traceability: Ensure it is easy to trace each sample from your published papers and raw data back to the holding institution, site publications, archaeological context and specialist reports. This makes it easier for others to use generated data, enhances reproducibility and prevents unnecessary repetitions of destructive sampling.
Change underlined for 5.3 no 2: “Traceability: Require that specimen or collection numbers for each sample are included along with references to the original site and specialist reports when available. This can be museum catalogue numbers or site specific numbers that can be linked back to the original excavation archive. Numbers assigned by laboratories that perform destructive analysis usually can’t be directly traced back to archaeological context so they are not sufficient.

Reviewer 2

1. Line 21: crustaceans replaced with arthropods.

2. Specimen scarcity is now emphasised through the underlined, added sentence in the following paragraph: A recent study of cats from archaeological sites in Denmark found 30 cat specimens spanning the time period from 1100 BCE to 1000 CE, opposed to a much higher number of 3485 specimens from the period 1000–1660 CE [33], illustrating the often uneven distribution of finds between time periods. The number of bones from a species found at a site can also be misleading as often only a fraction come from well dated contexts with clear taphonomic histories [34].”
3. Lines 135-8 and 143-5 changed to “However, these analyses are most fruitful when done on archaeofaunal collections which have already been analysed with traditional zooarchaeological methods [7,23,32,45]. Taphonomic processes often complicate the stratigraphy of archaeological sites which in turn makes the dating of individual animal bones and teeth challenging [34]. Since the interpretation of data generated by aDNA or isotope studies relies heavily on the dating of a sample and its archaeological context, careful sample selection, that takes confounding factors such as taphonomy and stratigraphy into account, is needed to produce meaningful results [18,23,34,45].”
4. Line 155-6: ‘species determination’ changed to ‘taxonomic determination’
5. Line 204-5: Changed to “Although sampling methods are constantly being improved to reduce sample damage and maximise data recovery, these efforts primarily focus on the human petrous and they are not always applicable to animal petrous bones due to anatomical differences [54,104,109,116].” and two more references added.
6. Added new point no 9 in 5,2: “**Saying no:** Sometimes holding institutions need to decline requests for destructive sampling e.g. when proposed projects can't meet sampling guidelines or the requested material is unsuitable for destructive analysis. Rejections should be given in a timely manner and state clearly why the request was declined. In some cases, alternative samples or sampling strategies can be suggested which would still allow a project to reach most of its goals.”
1. Line 305-6: Changed to “Acknowledgement: Require clear acknowledgment of the holding institution in all publications. The holding institution should receive copies of all publications that utilize samples or data derived from the institutions collection. This can help justify funding for curation and storage facilities.” Following this also added in 5.1 no 9 “**Share generated data:** Share raw data in public databases with clear meta-data for each sample that can be connected to publications and holding institutions. The holding institution should receive copies of all publications that utilize samples or data derived from the institutions collection.”
7. Reference Rainsford, O’Connor & Connelly 2016 added in several places where appropriate.

Other changes

1. Two references to newly published and relevant papers were added
 - a. Fox K, Hawks J. 2019 Use ancient remains more wisely. *Nature* 572, 581–583. (doi:10.1038/d41586-019-02516-5)
 - b. Bar-Oz G, Marom N, Pinhasi R. 2019 Technical Note: Taxonomic identification of petrosal bone morphology. *Bioarchaeology East* 13, 7.
2. Minor change in wording underlined and the adding of a reference in which had been accidentally removed in 5.1 no 10: “This can be done as part of the publication of positive results, by publishing a sampling report [e.g. 118] or in dedicated journals.”
3. In a few places we fixed made minor errors in grammar and spelling.

Kind regards,

Albína Hulda Pálsdóttir